# Molecular Characteristics and Functional Identification of a Key Alpha-Amylase-Encoding Gene *AMY11* in *Musa acuminata*

**DOI:** 10.3390/ijms25147832

**Published:** 2024-07-17

**Authors:** Peiguang Sun, Zhao Zhu, Zhiqiang Jin, Jianghui Xie, Hongxia Miao, Juhua Liu

**Affiliations:** 1National Key Laboratory of Tropical Crop Biological Breeding, Institute of Tropical Bioscience and Biotechnology and Sanya Research Institute, Chinese Academy of Tropical Agricultural Sciences, 4 Xueyuan Road, Haikou 571101, China; sunpeiguang888@sina.com (P.S.); hxmrain@163.com (Z.Z.); 18689846976@163.com (Z.J.); xiejianghui@itbb.org.cn (J.X.); 2College of Tropical Crops, Hainan University, 58 Renmin Avenue, Haikou 571100, China; 3Hainan Key Laboratory for Protection and Utilization of Tropical Bioresources, Hainan Institute for Tropical Agricultural Resources, Chinese Academy of Tropical Agricultural Sciences, Haikou 571101, China

**Keywords:** alpha-amylase, *cis*-acting elements, functional regulation, genome-wide identification, *Musa acuminata* L., starch degradation

## Abstract

Alpha-amylase (AMY) plays a significant role in regulating the growth, development, and postharvest quality formation in plants. Nevertheless, little is known about the genome-wide features, expression patterns, subcellular localization, and functional regulation of *AMY* genes (*MaAMYs*) in the common starchy banana (*Musa acuminata*). Twelve MaAMY proteins from the banana genome database were clustered into two groups and contained a conserved catalytic domain. These *MaAMYs* formed collinear pairs with the *AMY*s of maize and rice. Three tandem gene pairs were found within the *MaAMY*s and are indicative of putative gene duplication events. *Cis*-acting elements of the *MaAMY* promoters were found to be involved in phytohormone, development, and stress responses. Furthermore, *MaAMY02*, *08*, *09*, and *11* were actively expressed during fruit development and ripening. Specifically, *MaAMY11* showed the highest expression level at the middle and later stages of banana ripening. Subcellular localization showed that MaAMY02 and 11 were predominately found in the chloroplast, whereas MaAMY08 and 09 were primarily localized in the cytoplasm. Notably, transient attenuation of *MaAMY11* expression resulted in an obvious increase in the starch content of banana fruit, while a significant decrease in starch content was confirmed through the transient overexpression of *MaAMY11*. Together, these results reveal new insights into the structure, evolution, and expression patterns of the *MaAMY* family, affirming the functional role of *MaAMY11* in the starch degradation of banana fruit.

## 1. Introduction

Plant starch is the major storage carbohydrate in roots, leaves, tubers, seeds, and fruits [1,2]. Starch degradation in various organs not only provides energy for seed germination, seedling growth, and development in cereals [3,4,5], but also plays a crucial role in the postharvest quality formation, including sweetness, waxiness, and softening in banana, kiwifruit, and mango fruits [6,7,8]. The process of starch degradation is complex and involves the synergistic actions of many enzymes, such as α-amylase (AMY), β-amylase (BAM), α-glucosidase (GLU), starch phosphorylase (PHO), and limit dextrinase (LD) [8,9]. Among these enzymes, AMY is a glycosyl hydrolase with endoglycolytic activity on the α-1,4-glycosidic linkage in starch and is one of the key enzymes in converting starch into glucose and maltose via the combined action of BAM, GLU, and LD [3,10].

*AMYs* are encoded by a multi-gene family that has been screened in many plants through genome data analysis: three family members have been identified in kiwifruit (*Actinidia deliciosa*) and *Arabidopsis thaliana*; six in cassava (*Manihot esculenta*) [10]); seven in wheat (*Triticum aestivum*) [11]; ten in rice (*Oryza sativa*) [11] and barley (*Hordeum vulgare*) [12]; eleven in apple (*Malus domestica*) [13]. In vascular plants, AMYs can be divided into three subfamilies (AMY1, 2, and 3), each of which typically contains a conserved α-amylase catalytic domain (ACD, PF00128) [13]. Amy1 members have a signal peptide that activates the mobilization of starch in cereal endosperms [3]. Amy2 members have no signal peptide, and their function remains largely unknown despite reports of cytoplasmic localization [14]. However, Amy3 members have a large N-terminal extension of 400–500 amino acids in length, which contains tandem carbohydrate-binding modules and a predicted chloroplast transit peptide. The functional role of Amy3 is in leaf starch breakdown [10].

The expression patterns and functions of *AMYs* have been revealed in *A. thaliana* and cereals. In *A. thaliana*, *AMY1* was expressed at low levels in the leaf stalks and roots and was highly expressed in the leaves, stems, and flowers [15]. Upon stress treatment, the up-regulated expression of *AMY1* conferred high heat tolerance [15]. Among the ten rice *AMY*s, most were specifically expressed during the seed germination process and in the developing seed embryo [3]. In contrast to rice and *A. thaliana*, barley *AMY* expression was not found to be tissue-specific [16]. In wheat grains, the expression of a *late-maturity AMY* gene was induced by a cool temperature shock close to physiological maturity [17]. Moreover, a 20- to 230-fold increase in total AMY activity in mature grains was caused by the over-expression of wheat *AMY1* [18]. By contrast, the antisense suppression of an *AMY1* in the rice endosperm led to delayed germination [19]. In maize kernels, RNA interference (RNAi) with *AMY* gene expression in *Aspergillus flavus* led to decreased aflatoxin production [20]. The RNAi-mediated suppression of rice *AMY1A*, *1C*, *3A*, *3D*, and *3E* led to fewer chalky grains in ripening seeds under high-temperature conditions [21]. Together, the abovementioned reports imply that the expression of *AMYs* plays a crucial role in enhancing seed germination, grain development and maturation, and tolerance to stresses.

Around 70–80% of the dry weight of unripe banana fruits is composed of starch. During the ripening process, this starch is rapidly converted into soluble sugars by a series of starch-degrading enzymes [8,22]. Early research by Bassinello et al. [23] reported that AMY activity might be involved in the early stage of starch degradation during banana fruit ripening. Subsequently, Junior et al. [24] found that, among several hydrolytic enzymes, AMY is the only enzyme able to act upon intact granules. More recently, it was shown that the transcription factor BEL1-LIKE HOMEODOMAIN 1 interacted with the *MaAMY3* gene promoter to accelerate softening and ripening in banana fruit [25]. However, the genome-wide systematic screening of *MaAMY*s associated with the starch degradation in banana fruit remains to be further researched. The genome-wide characteristics, subcellular localization, expression features, and functional regulation of the key *MaAMY*s are still unclear. In the current study, we identified and characterized 12 distinct *MaAMY* members in the banana A genome database and evaluated the evolutional features as well as the transcriptome and spatio-temporal expression patterns. Furthermore, the subcellular localization of four expressed proteins, namely, MaAMY02, 08, 09, and 11, was analyzed in *Nicotiana benthamiana* leaf cells. The functional role of the most highly expressed protein, *MaAMY11*, during ripening was determined by transient silencing and transient overexpression in banana fruit. Together, our findings enhance our understanding of the function of *MaAMY11* in fruit starch degradation and provide key target genes for the genetic improvement of quality formation in common starchy bananas and other crops.

## 2. Results

### 2.1. Identification and Physicochemical Characterization of MaAMY Proteins in Banana

All AMY protein sequences were screened from the banana A genome database [26]. A total of 12 AMY proteins, named MaAMY01–MaAMY12 based on their order on the chromosomes, were identified and possessed a conserved ACD (PF00128). The physicochemical characteristics of the MaAMYs were systematically analyzed (Appendix A). The coding sequences (CDSs) and amino acid (aa) lengths were in the range of 285 (MaAMY01)–2841 bp (MaAMY11) and 94 (MaAMY01)–948 aa (MaAMY11), respectively. C, H, N, O, and S constituted the main elemental composition of their amino acids. Their isoelectric points and molecular weights were in the ranges of 5.20 (MaAMY12)–9.23 (MaAMY03) and 10.89 (MaAMY01)–106.66 kDa (MaAMY11), respectively. Their aliphatic indexes ranged from 67.10 (MaAMY04) to 93.19 (MaAMY01), indicating that most were thermostable. All proteins except MaAMY02, 03, 07, 08, 09, and 10 were unstable (>40), and all except MaAMY01 were hydrophilic (<0), according to the instability calculations and grand average of hydropathicity (GRAVY) scores. MaAMY01, 02, 03, and 11 were localized in the chloroplast; MaAMY04, 05, and 06 were localized in the nucleus; MaAMY07, 08, 09, and 12 were localized in the cytoplasm; and MaAMY10 was localized in the vacuole (Appendix A), implying that they may be involved in different biological processes.

### 2.2. Motifs, Gene Structure, and Multiple Sequence Alignments of MaAMY Family Members

Within the banana genome, 12 MaAMY proteins were classified into groups I and II; MaAMY01, 02, 03, 07, 08, 09, 10, and 12 were in group I; MaAMY04, 05, 06, and 11 were in group II (Figure 1A). The MaAMY members in group I contained motifs 2–12, while the MaAMYs in group II contained conserved motif 11 (Figure 1B). The exon–intron structure of the *MaAMY*s was further analyzed (Figure 1C). Group I had relatively few introns, with 1–4 introns found in *MaAMY01*, *07*, *08*, *09*, *10*, and *12*. In group II, *MaAMY04*, *05*, *06*, and *11* had 12 introns (Figure 1C), implying that the *MaAMYs* in group II have similar motifs and exon–intron structures. Multiple sequence alignments showed that each MaAMY protein contained an ACD (Figure 1D and Appendix A), which is a hallmark of AMY proteins [3,10]. In addition, an alpha-amylase C-terminal domain or a signal peptide was found in MaAMY02, 04, 05, 06, 08, 09, 10, 11, and 12 (Appendix A). Interestingly, MaAMY11 has the longest amino acid sequence in the gene families compared with bananas and other plants (Figure 1D and Appendix A).

### 2.3. Evolutionary Relationships of AMYs from Banana and Other Plant Species

To investigate the evolutionary relationships of the MaAMYs, a total of 108 AMY protein sequences from *M. acuminata*, *A. thaliana*, *Brachypodium distachyon*, *Capsicum annuum*, *H. vulgare*, *O. sativa*, *Panicum hallii*, *Seturia italica*, *Solanum tuberosum*, *Solanum lycopersicum*, *T. aestivum*, *Sorghum bicolor*, and *Zea mays* were used to construct a tree (Figure 2A). All of the accession numbers for these proteins are provided in Appendix A. Groups I, II, and III were present in the classification analysis of these proteins. The majority of banana MaAMYs, including eight MaAMY (MaAMY01, 02, 03, 07, 08, 09, 10, and 12) members, were found in group III. Four members, namely, MaAMY04, 05, 06, and 11, were found in group II. However, zero MaAMY members in group I were detected in the banana genome, in contrast to monocot plants such as *T. aestivum* and *Z. mays*, in which 1–2 group I members have been found (Figure 2A), implying that the banana MaAMYs have undergone different genome evolutionary processes compared with other monocot plants. Furthermore, with the exception of pepper, a Ka/Ks ratio of less than one was found in bananas and other plant species (Figure 2B), indicating that strong purifying selection had occurred in the *MaAMYs*.

### 2.4. Chromosomal Localization and Tandem Duplication of MaAMYs and Cis-acting Element Analyses of MaAMY Gene Promoters

Twelve *MaAMY* genes were mapped on six chromosomes in bananas. These included *MaAMY01*, *02*, and *03* on Chr 3; *MaAMY04*, *05*, and *06* on Chr 4; *MaAMY07*, *08*, and *09* on Chr 5; *MaAMY10* on Chr 7; *MaAMY11* on Chr 8; and *MaAMY12* on Chr 10 (Figure 3A). Moreover, Figure 3A revealed the discovery of three tandem gene pairs (*MaAMY01*, *02*, and *03*; *MaAMY04*, *05*, and *06*; *MaAMY07* and *08*) which may have undergone gene tandem duplication events.

A promoter region including a 2000 bp upstream sequence of each *MaAMY* gene was selected; promoter structure comparisons and the *cis*-elements were confirmed by the database (http://www.fruitfly.org/seq_tools/promoter.html, accessed on 20 June 2023) and PlantCARE software (http://bioinformatics.psb.ugent.be/webtools/plantcare/html/, accessed on 20 June 2023), respectively. Notably, these promoter regions did not contain 5′ untranslated region (UTR) introns, but all of the promoters contained transcription start sites (TSSs). The location information for the TSSs is provided in Appendix A. The typical core sequences, namely, the TATA-box (9–144) and CAAT-box (8–20), were enriched in each *MaAMY* promoter region (Figure 3B). *Cis*-phytohormone-responsive, development- and growth-related, stress-responsive, light-responsive, or circadian-control elements were found in most *MaAMY* promoters (Figure 3B). The phytohormone-related elements ABRE, ARE, CGTCA, and GARE were identified in the *MaAMY11*, *09*, *07*, and *05* promoters. Abiotic stress-related elements, including LTR and MBS, were identified in the *MaAMY07*, *04*, and *01* promoters. Box4, G-box, GT1-box, and I-box light-responsive-related elements were identified in the *MaAMY01* and *06* promoters. The development-related elements (MRE and O_2_-site) existed in the *MaAMY11*, *09*, *04*, or *02* promoters. 

### 2.5. Intergenomic and Intragenomic Collinearity Analysis

Different linear relationships among banana chromosomes were revealed. There was collinearity between *MaAMY10* and *MaAMY12* in the intrachromosomal regions (Figure 4A). The intragenomic collinearity of the *AMY*s among banana, tomato, maize, rice, and *A. thaliana* was further analyzed. Three *MaAMY* members were collinear with three rice *OsAMY*s and three maize *ZmAMY*s (Figure 4B). In contrast, *MaAMYs* did not exhibit collinearity with *A. thaliana AtAMYs* and tomato *SlAMYs* (Figure 4C). Notably, the *AMY*s were mainly collinear between banana and rice, and between banana and maize, based on the comparative genomic analysis, and these duplicated genes may have been lost or altered in different species due to evolutionary processes.

### 2.6. MaAMY11 Exhibits Higher Expression than Other MaAMYs in the Fruit or during Ripening

Transcriptome data (SRX3938715, SRX3938722, SRX3938708, SRX3938709, SRX3938706, SRX3938707, and SRX3938704) from various tissues and fruits at different developmental stages were used to analyze *MaAMY* gene expression patterns (Appendix A). Five *MaAMYs*, including *MaAMY02*, *08*, *09*, *10*, and *11*, were expressed (reads per kilobase pair of transcripts per million reads, RPKM > 7.9) in at least two of the tested tissues at 80 days after the emergence of the inflorescence from the pseudostem (DAF) from ‘Cavendish’ (*M. acuminata* AAA genotype). However, the expression of *MaAMY11* was high in both the leaves and fruits (RPKM > 79.6). Moreover, *MaAMY01*, *03*, *04*, *05*, *06*, *07*, and *12* were not expressed in different organs (Figure 5A).

The temporal expression patterns of the *MaAMYs* were analyzed in fruit sampled at 0, 20, and 80 DAF and 8 and 14 days after harvest (DPH) (Figure 5B, Appendix A). Except for *MaAMY01*, *03*, *04*, *05*, *06*, *07*, *10*, and *12*, four *MaAMY* genes (*MaAMY02*, *08*, *09*, and *11*) were expressed (RPKM > 1.9) during banana development and ripening. Among the expressed *MaAMYs*, *MaAMY11* showed the highest transcriptional accumulation (82.7 < FPKM < 1426.7) at the middle and later stages of banana fruit ripening.

The spatio-temporal expression characteristics of the four expressed genes (*MaMAY02*, *08*, *09*, and *11*) were further identified using quantitative real-time polymerase chain reaction (qRT-PCR). Following normalization, the relative expression of each gene in different organs or at different temporal stages was verified through comparison to the root and at 0 DAF, with the relative expression of each gene in the root or at 0 DAF considered “1”. The qRT-PCR results showed that all four *MaAMYs* exhibited better agreement with the RNA-seq data (Figure 5C–J). The correlation coefficients exceeded 0.9730 to 0.9998 for both the RNA-seq and qRT-PCR data in the different organs and at various temporal stages, respectively, implying that the RNA-seq data were reliable. Moreover, a higher abundance of *MaAMY11* than other *MaAMYs* in the fruits or during banana fruit ripening was revealed by both the transcriptome data and qRT-PCR data, suggesting that *MaAMY11* may play a crucial role in driving fruit ripening.

### 2.7. Co-Localization of Four Expressed MaAMY Proteins

The open reading frames (ORFs) of the abovementioned expressed *MaAMY02*, *08*, *09*, and *11* were inserted into pCAMBIA1302-GFP vectors, respectively. These four recombinant constructs were transiently expressed into *N. benthamiana* leaves via agroinfiltration, from which four fusion proteins were generated, including MaAMY02-GFP, MaAMY08-GFP, MaAMY09-GFP, and MaAMY11-GFP. The subcellular co-localization results showed that the MaAMY02-GFP and MaAMY11-GFP proteins were localized in the chloroplast with a co-localized chloroplast marker (red fluorescent protein, RFP). The MaAMY08-GFP and MaAMY09-GFP proteins were mainly localized in the cytoplasm with a co-localized cytoplasm marker (RFP). By contrast, in the GFP-positive control (PC), fluorescence distribution across the entire cell was observed in the *N. benthamiana* leaf cells (Figure 6).

### 2.8. Banana MaAMY11 Plays a Crucial Role in Fruit Starch Degradation

The function of *MaAMY11*, which had the highest expression level during banana ripening, was evaluated. As shown in Figure 7A, transient silencing of *MaAMY11* expression led to darker staining by iodine–potassium–iodide (I_2_-KI) in banana fruit discs compared to the controls. In the transiently silenced fruit discs, the expression level of *MaAMY11* was obviously decreased (Figure 7B), but the contents of total starch, amylose, and amylopectin were obviously increased by 9.49%, 3.76%, and 5.73%, respectively, compared with the empty vector (Figure 7C–E). By contrast, lighter staining by I_2_-KI was exhibited in the banana fruit discs by transiently overexpressing *MaAMY11* (Figure 7F). The expression level of *MaAMY11* was obviously increased in overexpressing fruit discs (Figure 7G), whereas the total starch, amylose, and amylopectin contents were significantly decreased by approximately 31.77%, 6.81%, and 24.97%, respectively (Figure 7H–J).

## 3. Discussion

Starch is an important fundamental substance for yield and quality formation in banana [26,27]. In our previous research, a series of genes associated with banana starch biosynthesis were identified [28,29,30,31]. Recently, some genes or transcription factors involved in banana starch degradation, including *BAM9b* [32], *PHO1* [8], MYB3 [33], bHLH6 [34], MYB16L [35], APETALA2 (AP2a) [36], ERF12 [37], NAC029 [38], BEL1 [25], CCCH33-like2 [39], and C2H2 [40], were reported. AMY is an important enzyme that catalyzes the hydrolysis of fruit starch degradation [41], and the genome-wide characteristics of the *MaAMY* genes in banana have not yet been assessed. Herein, 12 members (MaAMY01–MaAMY12) were identified in the banana A genome database (Figure 1 and Appendix A). An ACD was found in all MaAMYs, and these *AMY* gene members have relatively conservative motifs and exon–intron structures (Figure 1). The finding corroborates other reports in plants, such as rice [3], wheat [9], and cassava [10]. Notably, we found that MaAMY11 has the longest amino acid sequence compared with other plants (Appendix A). However, further research is necessary to determine whether MaAMY11 has a special function in multiple biological processes.

We found that banana MaAMY proteins were categorized into two groups (groups II and III), with group I not present. This differs from *T. aestivum* and *Z. mays* (Figure 2A) and vascular plants [3,10,12,14], suggesting that the phylogenetic evolution of banana *MaAMYs* may be different from *T. aestivum*, *Z. mays*, or vascular plants. It may also be because starch biosynthesis and degradation in bananas are completed in the same generation, whereas starch biosynthesis in other monocot plants such as *T. aestivum* occurs in the first generation, with starch degradation completed in the next generation. In addition, we found that MaAMYs in group II demonstrated a close evolutionary relationship to orthologs AtAMY03 and OsAMY04 in *A. thaliana* and *O. sativa*. The closest orthologs to MaAMYs in group III include TaAMY06 and 13 in *T. aestivum* (Figure 2A). A Ka/Ks ratio of less than one was detected in all MaAMYs (Figure 2B), implying that a strong purifying selection may have occurred during the evolution of the banana MaAMYs [42]. Gene duplication is an important mechanism for acquiring new genes during evolution [43]. The present analysis found that banana *MaAMY* expansion was likely the result of tandem duplication (Figure 3A). An unbalanced distribution of *MaAMY*s on the chromosomes, with chromosomes 3, 4, and 5 containing three-fourths of the entire *MaAMY* gene family, was also found. Furthermore, the collinearity analysis identified intrachromosomal collinear pairs between *MaAMY10* and *MaAMY12* within the banana genome (Figure 4A), three collinear *AMY* gene pairs between banana and maize, and three gene pairs between banana and rice (Figure 4B). This suggests that the duplicated *AMY*s might have undergone divergence during the evolutionary process. Similar results have been reported in the synthetic analysis of *AMY*s from cassava [10].

The *AMY* promoters contain several *cis*-acting elements that are associated with the growth, development and maturation, and abiotic and biotic stresses in plants [11,44]. In rice, three elements include a G-box-related element, the amylase element, and a CGACG element that regulate the high-level expression of rice *Amy3D* during seedling development [45]. Loss- and gain-of-function studies have reported that the sugar response sequence of the rice *AMY3* promoter serves as a transcriptional enhancer under sugar starvation conditions [46]. In barley and wheat, *AMY1* promoters contain a GARE, pyrimidine box, and TATCCAT/C box [11]. In *Camellia sinensis*, *cis*-acting elements of *CsAMY* promoters determining the quality during the postharvest processing of tea leaves were analyzed [44]. In *MaAMY* promoters, the phytohormone-related ABRE, ARE, CGTCA, and GARE, the stress-related LTR and MBS, and the development- and growth-related elements (MRE and O2-site) have been found. *MaAMY11* was particularly enriched in phytohormone- and development-related elements (Figure 3B), which may play an important role in phytohormone- and development-related biological processes. In addition, a large number of conserved *cis*-acting elements, including the TATA-box and CAAT-box, were found in *MaAMY* gene promoter regions, which were consistent with the characteristics of the *AMY* gene promoters in rice [45] and cassava [10].

*AMYs* are expressed during seed germination and fruit development in several plant species, such as rice [3], wheat [47], potato [48], and kiwifruit [41]. Among the ten rice *AMYs*, most are highly expressed during the seed germination process and in the developing seed embryo [3,49]. In wheat, *AMYs* were significantly highly expressed at 6 days after anthesis, suggesting that AMY likely catalyzes the hydrolysis of starch granules [47]. In kiwifruit, the *AdAMY1* expression was obviously induced by ethylene and was positively correlated with starch degradation [41]. In the present study, *MaAMY02*, *08*, *09*, and *11* were expressed during fruit development, and *MaAMY11* was the only gene that was highly expressed during banana fruit ripening (Figure 5), implying its involvement in fruit ripening. Furthermore, the MaAMY02 and 11 proteins were localized in the chloroplast, which is consistent with previous reports in rice *AMYI-1* [19] and *A. thaliana* AtAMY3 [48]. The MaAMY08 and 09 proteins were also found to be localized in the cytoplasm (Figure 6). This finding is consistent with a previous report in potato StAMY23 [50].

Suppressed or overexpressed *AMYs* influence the starch metabolism in rice, potatoes, and wheat [4,19,48]. In rice, the suppressed expression of *AMYI-1* resulted in an obvious increase in starch accumulation in the young leaf; the overexpression of *AMYI-1*, *1A*, *3C*, and *3D* resulted in a reduction in the starch content in the leaves or in the developing endosperm [19,49]. Silencing potato *StAMY23* expression led to higher phytoglycogen and lower resistant starch accumulation in tubers and delayed tuber sprouting [48,50]. In wheat, the overexpression of *TaAMY2* resulted in an absence of dormancy in the ripened grain [4], whereas the overexpression of *TaAMY3* led to an increase in the total AMY activity, but the increased activity did not affect the starch content or the composition of the dry grain [51]. Moreover, in *A. thaliana*, double-knock-out *AtAMYs* (*AtAMY1*, *2*, and *3*) combinations and a triple-knock-out mutant did not have a significant impact on starch degradation in the leaves [52]. We were interested in whether the high expression of *MaAMY11* during banana ripening has an impact on fruit starch degradation. Herein, the transient silencing of *MaAMY11* expression resulted in an obvious increase in the total starch content, followed by lighter I_2_-KI staining, which corroborates previous reports in rice [19] and potato [48]. However, this result differs from reports in dry wheat grain [51] and *A. thaliana* leaf [50]. Furthermore, the transient overexpression of *MaAMY11* resulted in an obvious decrease in the content of total starch in the banana fruit. This finding agrees with previous assumptions that AMY may be important for fruit starch degradation in bananas [53]. In the current study, direct evidence supports that *MaAMY11* plays an important role in the starch degradation of banana fruit by transient silencing or transient overexpression.

## 4. Materials and Methods

### 4.1. Plant Materials

The banana cultivar ‘Cavendish’ (*M. acuminata* AAA genotype) was obtained from the Banana Germplasm Nursery (19°N, 109°E) located in the Chinese Academy of Tropical Agricultural Sciences, Danzhou City, Hainan Province, China. The different organs (roots, leaves, and fruit) were harvested at 80 days after the emergence of the inflorescence from the pseudostem (DAF) for spatial expression analysis. Following fruit development and ripening, pulps at 0, 20, and 80 DAF, and 8 and 14 days postharvest (DPH), were selected for temporal expression analysis. Three replicates were conducted for these expression studies.

### 4.2. Genome-Wide Identification of Banana AMY Family Genes

The *AMY* family members were identified based on the banana genome database [26]. AMY amino acid sequences were defined by the InterPro database [54]. An HMM profile was analyzed by the Pfam database. Twelve *MaAMY* members were annotated, and their ACDs (PF00128) were confirmed using the Conserved Domain Database and SMART software v.8.0. The physicochemical properties of *MaAMY1*–*MaAMY12* proteins were annotated by the ExPASy database. The secondary structures and subcellular localizations of the MaAMY proteins were clarified with SOPMA and the Cell-PLoc 2.0 web-based tool, respectively. Appendix A presents all of the bioinformatic analytical websites, databases, and software used in this study.

### 4.3. Motifs, Structures, and Multiple Sequence Alignment

To identify conserved motifs within the amino acid sequences, we used Motif Elicitation analysis (MEME) and set the maximum motif count to 12 [55,56]. GSDS v. 2.0 was used to elucidate the exon–intron structures. In addition, for a comprehensive analysis, we performed multiple sequence alignments on the *MaAMY* gene family using MEGA X v. 10.1.1 software. The conserved motifs, gene structure, and multiple sequence alignment characteristics were visualized using TBtools v.1.0 [57].

### 4.4. Phylogenetic Tree and AMY Gene Family Evolutionary Selection Pressure Analysis

The annotation data for the genomes of various plant species, including *M. acuminata*, *B. distachyon*, *A. thaliana*, *C. annuum*, *O. sativa*, *H. vulgare*, *P. hallii*, *S. italica*, *S. bicolor*, *S. lycopersicum*, *S. tuberosum*, *Z. mays*, and *T. aestivum*, were sourced using the banana A genome, phytozome v13 database, and ensembl plants database (Appendix A). To identify relevant *AMY* genes across these species, we utilized the Basic Local Alignment Search Tool (BLAST) within TBtools v.1.0 [54]. Genes exhibiting an e-value of less than 0.00005 were deemed candidate genes [54].

The sequences and the presence of the ACD were verified using the NCBI’s CD-Search tool and SMART software v.1. A phylogenetic tree of AMY amino acid sequences from bananas and other species was constructed using MEGA X v. 10.1.1 with 1000 bootstrap replications and the neighbor-joining (NJ) method, following which the phylogenetic tree was enhanced using the iTOL website. In addition, we used the itol.toolkit package in R for further modifications [58]. We extracted the CDS sequences of the *AMYs* from the various species genomes using the phytozome v13 database. The ratio of nonsynonymous (Ka) and synonymous (Ks) substitutions for these CDS sequences was calculated by KaKs Calculator v. 1.2. Finally, the results were visually analyzed and presented using the ggplot2 package in R [59].

### 4.5. Chromosome Localization, Gene Duplication and Collinearity, and Promoter Cis-Acting Element Analyses

The chromosomal localization of the *MaAMY*s was analyzed using Mapchart v. 2.3. Gene duplication and collinearity were analyzed using MCS-canX [60]. Genome collinearity was visualized using Circos software v. 0.69-9. The specific loci of the *MaAMY*s, as well as the 2000 bp upstream regions preceding them, were designated as promoter regions. The *cis*-acting elements of these promoter regions were predicted by the PlantCARE website [29]. TBtools software was used to determine the precise locations of these elements, and R software v. R-4.4.1 was used to construct a heatmap for quantifying the occurrence of each gene element [37].

### 4.6. Transcriptomic Analysis

RNA was extracted from the roots, leaves, fruit, and pulps of bananas at different developmental (0, 20, and 80 DAF) and ripening stages (8 and 14 DPH) using an RNAprep Pure Plant Kit (supplied by Tiangen, Beijing, China). An Illumina GAII platform was used for deep sequencing (Illumina, Inc., San Diego, CA, USA). Low-quality reads and adapter sequences were eliminated by the FastQC software and FASTX toolkit, respectively. Transcriptome assemblies were generated by Cufflinks. The gene expression levels were calculated based on the RPKM value. The DESeq package was used to screen the differentially expressed genes (DEGs). Two technical replicates and three biological replicates were evaluated in the sequencing process. According to the RPKM value of the *MaAMYs*, a heatmap was constructed using MeV 4.9.0 software.

### 4.7. Quantitative Real-Time Polymerase Chain Reaction and Statistical Analysis

The relative expression levels of the *MaAMYs* in roots, leaves, and fruit at different development stages and different ripening stages were performed with a SYBR^®^ Premix Ex Taq™ kit (TaKaRa, Shiga, Japan) on a qRT-PCR system (qTOWER3G, Analytik, Jena, Germany). The internal controls were *Actin* (EF672732) and *UBQ2* (HQ853254). Appendix A provides the primer sequence information. The expression levels of the *MaAMYs* relative to *UBQ2* and *Actin* were estimated by the 2^−ΔΔCT^ method [61]. Each sample included three replicates.

### 4.8. Transient Silencing or Transient Overexpression of MaAMY11 in Banana Fruit

The transient silencing vector pTRV2-*MaAMY11* was constructed using the *Eco*R I and *Kpn* I enzymes (Appendix A). The transient overexpression vector pCAMBIA3300-*MaAMY11* was constructed using the *Kpn* I and *Sal* I enzymes (Appendix A). The primer sequences are provided in Appendix A. The constructed pCAMBIA3300-*MaAMY11* and pTRV2-*MaAMY11* plasmids were then transferred into the GV3101 strain. Surface-sterilized banana fruit discs at 80 DAF were infiltrated with Agrobacterium (OD_600_ = 0.6) and placed on a Murashige and Skoog medium, incubated at 30°C for 3 d, and then stained with I_2_-KI solution [8,31]. The expression level of *MaAMY11* was detected in the transiently silenced or transiently overexpressed fruit slices. Total starch, amylose, and amylopectin contents were detected using the method reported by Miao et al. [8,31]. The experiment used triplicate biological replicates.

## Figures and Tables

**Figure 1 ijms-25-07832-f001:**
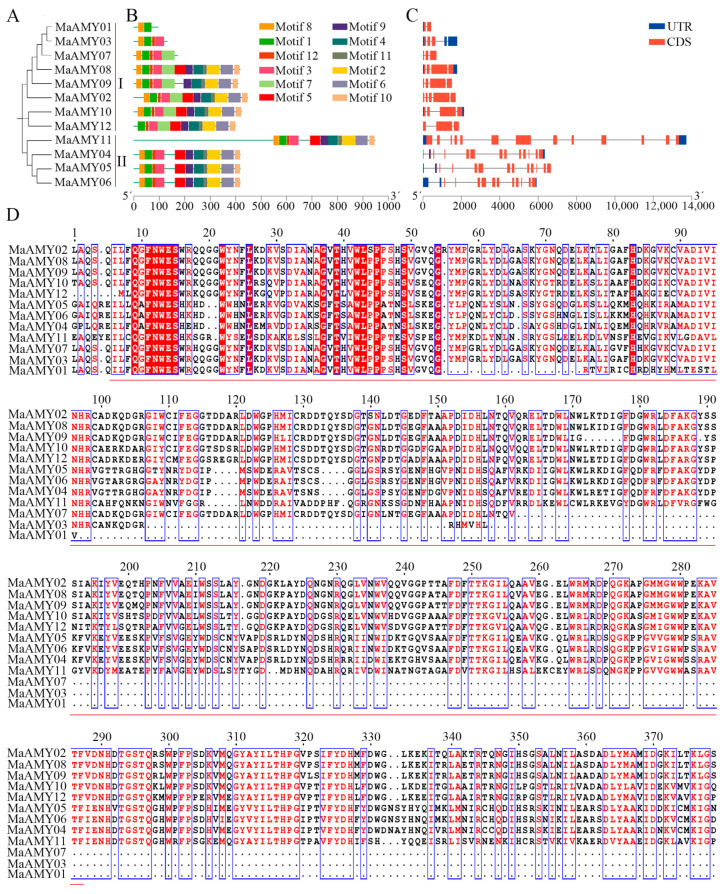
Bioinformatics analyses of the *MaAMY* gene family. (**A**) Phylogenetic evolution of the MaAMY protein family. UTR represents untranslated region. CDS represents coding sequence. (**B**) Motif analysis of the *MaAMY* gene family. (**C**) Gene structure analysis of the *MaAMY* gene family. (**D**) Multiple sequence alignment of *MaAMY* family amino acids, where red letters and blue boxes show the highly conservative positions and black letters indicate less conserved positions. Red line represents the α-amylase catalytic domain.

**Figure 2 ijms-25-07832-f002:**
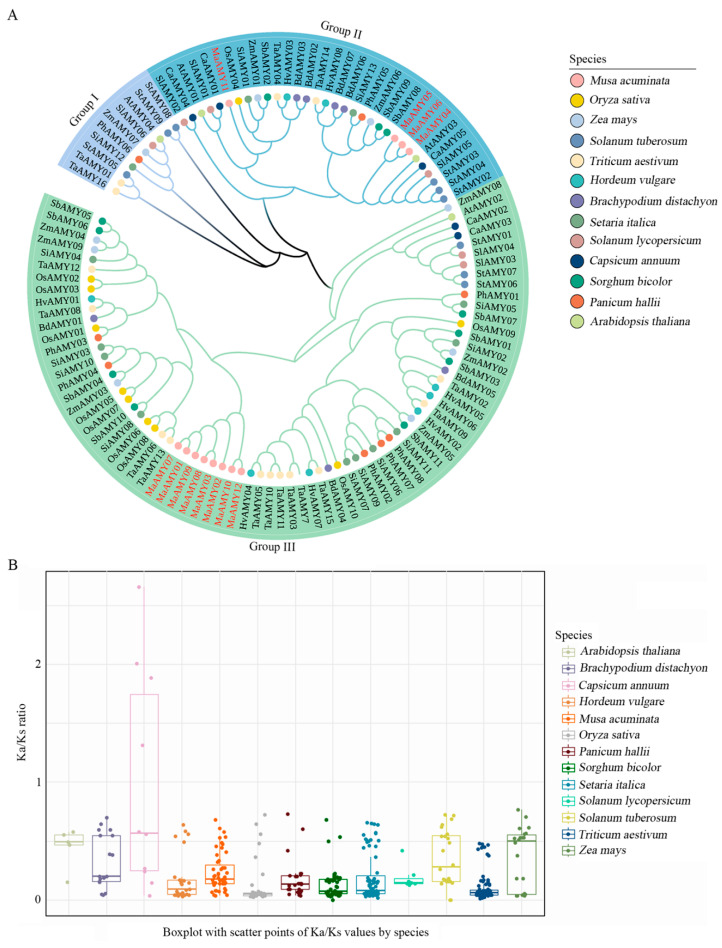
Phylogenetic and Ka/Ks ratio analysis of 108 AMY proteins from banana and 11 other plant species. The phylogenetic tree was constructed with the MEGA X v. 10.1.1 software. (**A**) The AMYs were divided into three groups, including group I, II, and III, based on the amino acid sequences of the AMYs in banana and other plants. The MaAMYs, OsAMYs, ZmAMYs, StAMYs, TaAMYs, HvAMYs, BdAMYs, SiAMYs, SlAMYs, CaAMYs, SbAMYs, PhAMYs, and AtAMYs, represent AMYs from *Musa acuminata*, *Oryza sativa*, *Zea mays*, *Solanum tuberosum*, *Triticum aestivum*, *Hordeum vulgare*, *Brachypodium distachyon*, *Seturia italica*, *Solanum lycopersicum*, *Capsicum annuum*, *Sorghum bicolor*, *Panicum hallii*, and *Arabidopsis thaliana*, respectively. (**B**) Boxplot with scatter points of Ka/Ks values by species.

**Figure 3 ijms-25-07832-f003:**
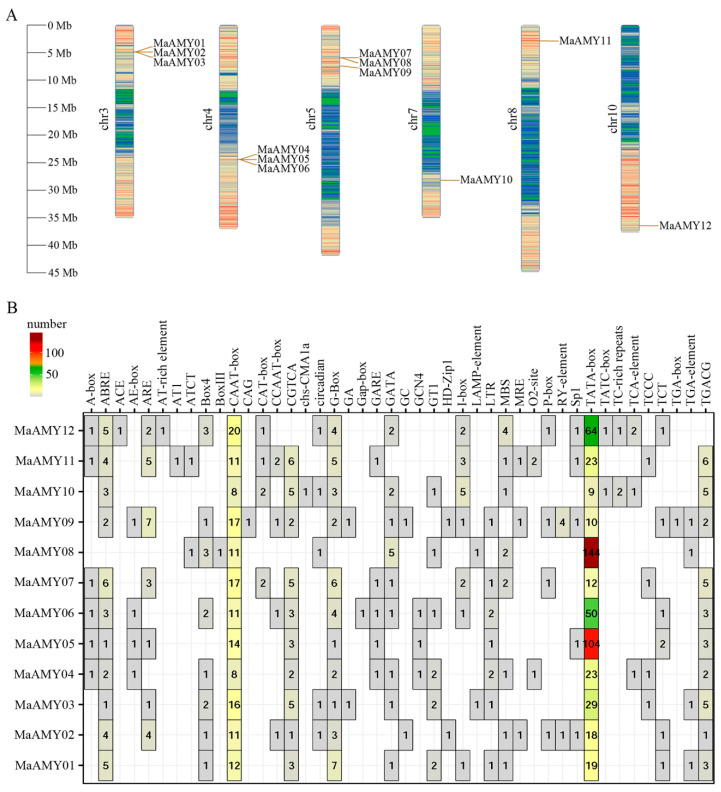
Chromosomal distribution and *cis*-acting element analysis of *MaAMY* family. (**A**) Chromosomal localization of 12 *MaAMY* gene family members. Green, blue, and orange colors represent low, medium, and high gene density in the region of the chromosome, respectively. (**B**) Promoter *cis*-acting elements of *MaAMY* genes; the number represents the number of *cis*-acting element in each *MaAMY* promoter.

**Figure 4 ijms-25-07832-f004:**
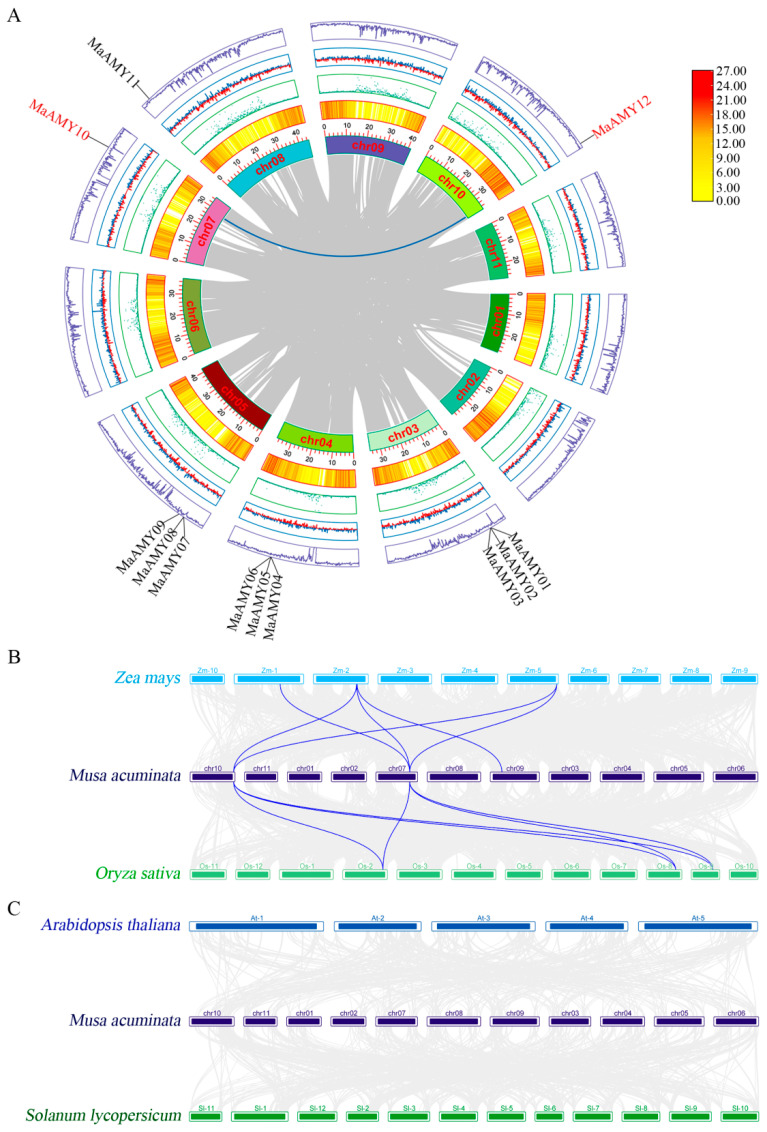
Intergenomic and intragenomic synteny relationship of the *AMY* genes among banana (*Musa acuminata*), tomato (*Solanum lycopersicum*), maize (*Zea mays*), Arabidopsis (*Arabidopsis thaliana*), and rice (*Oryza sativa*). (**A**) Intergenomic synteny relationship between the *MaAMY* genes in the banana genome. Purple lines indicate the GC content. Red lines indicate the GC Skew. Green dots indicate the Nratio. Gray lines indicate the interchromosomal collinear relationship in the banana A genome. Blue lines indicate the intrachromosomal collinearity between *MaAMY10* and *MaAMY12*. (**B**) Collinear *AMY* family gene pairs between tomato and banana and between banana and maize. (**C**) Collinear *AMY* family gene pairs between *A. thaliana* and banana and between banana and rice. Colored lines highlight the collinear gene pair.

**Figure 5 ijms-25-07832-f005:**
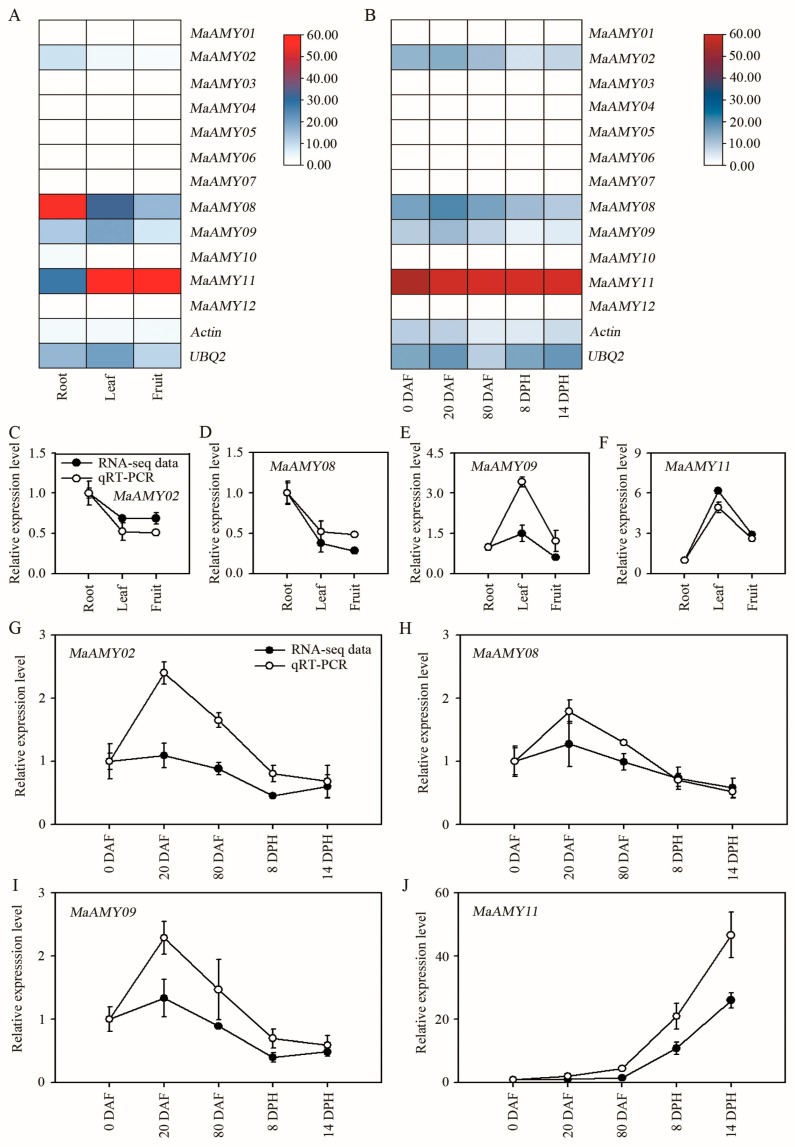
Expression profiles of *MaAMY* family genes in the different organs and different developmental stages by banana transcriptome and qRT-PCR. (**A**,**B**) Expression of *MaAMYs* in different organs and during different stages of banana fruit development and ripening. The heat map with clustering was created based on the FPKM value of the *MaAMYs*. Differences in gene expression changes are shown in color in the blue–red scale. (**C**–**F**) Expression of *MaAMY02*, *08*, *09*, and *11* in different tissues. (**G**–**J**) Expression of *MaAMY02*, *08*, *09*, and *11* at different stages of fruit development and ripening. Data are presented as means ± standard deviations, *n* = 3 biological replicates.

**Figure 6 ijms-25-07832-f006:**
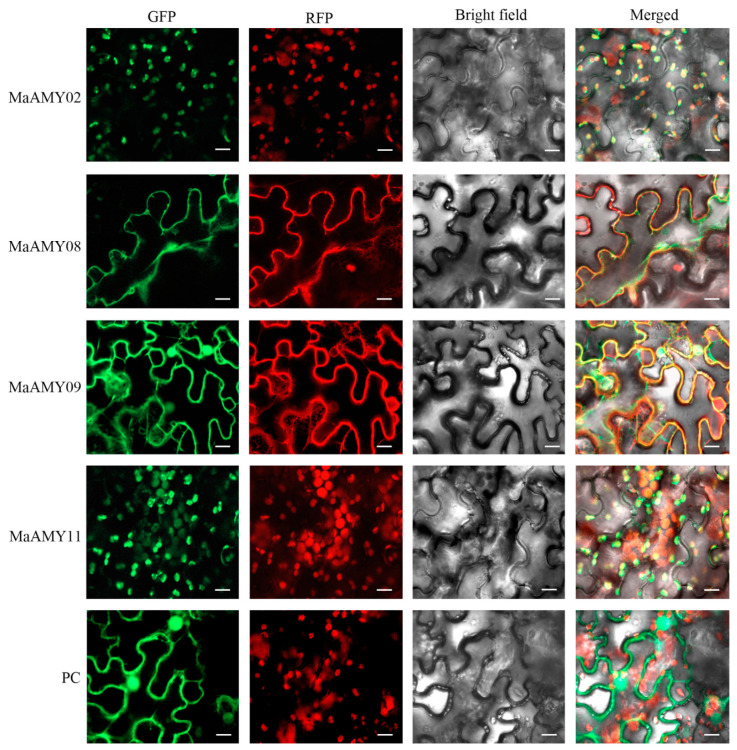
Co-localization of MaAMY02, 08, 09, and 11 proteins. The green fluorescent protein (GFP) fluorescence is represented by green, whereas red fluorescent indicates red fluorescent proteins (RFPs). A composite image was created by merging the GFP and RFP fluorescence images, represented under “merge”. PC represented the positive control (pCAMBIA1302-GFP plasmid). Scale bars = 10 μm.

**Figure 7 ijms-25-07832-f007:**
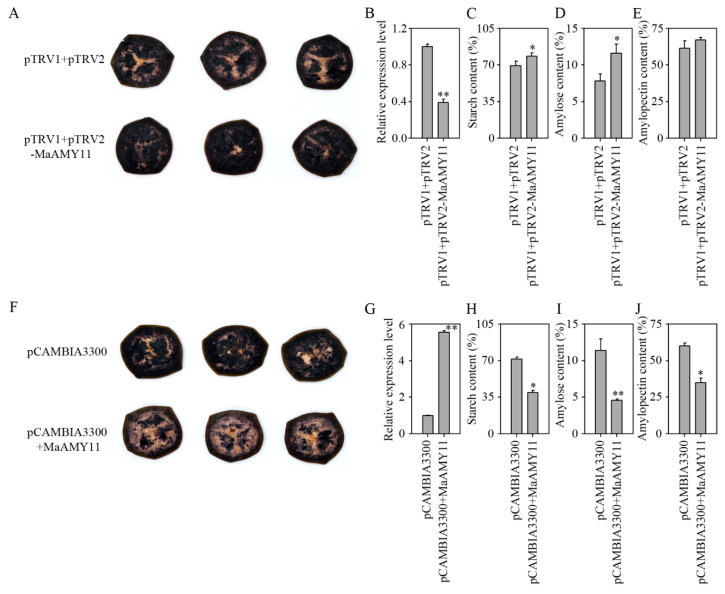
Transient suppression and overexpression of the *MaAMY11* gene in banana fruit discs. (**A**) I_2_-KI staining indicative of *MaAMY11* suppression in banana fruit discs. (**B**) Expression level of *MaAMY11* in *MaAMY11* suppression in banana fruit discs. (**C**–**E**) Variation in total starch, amylose, and amylopectin contents following *MaAMY11* suppression in banana fruit discs. (**F**) I_2_-KI staining of *MaAMY11*-overexpressing banana fruit discs. (**G**) Expression level of *MaAMY11* in *MaAMY11* overexpression in banana fruit discs. (**H**–**J**) Change in total starch, amylose, and amylopectin contents following *MaAMY11* overexpression in banana fruit discs. Triplicate replicates were tested, and the asterisks denote statistically significant differences compared to pTRV1 + pTRV2 (empty vector control) or pCAMBIA3300 (empty vector control) (*, *p* < 0.05; **, *p* < 0.01).

## Data Availability

No new data were created or analyzed in this study. Data sharing is not applicable to this article.

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
