# Peer review of "Molecular Characteristics and Functional Identification of a Key Alpha-Amylase-Encoding Gene *AMY11* in *Musa acuminata"

_ijms, 2024, doi:10.3390/ijms25147832_

Round 1
Reviewer 1 Report
Comments and Suggestions for Authors
The present study titled “Molecular characteristics and functional identification of the key alpha-amylase-encoding gene AMY11 in banana fruit” is presented in detailed well-elaborated research. In this paper, the alpha-amylase-encoding gene was systematically studied, and AMY11 was deeply analyzed and functionally identified. There are many mistakes in the article that need to be corrected and solved by the author, my comments related to improving the article are as follows:
1. In figure 1c, The genetic difference between MaAMY11 and other families is very large, so I think it is necessary for the author to correct this gene and redraw it. And the green squares in Figure 1c should refer to the UTR region rather than upstream and downstream.
2. In line 136-137 and 395-397 , species names should be italicized.
3. In figure 5, The clustering sequence of genes in the heat map of family gene expression is suggested to follow the sequence of evolutionary tree.
4. Why did the authors select only four genes, MaAMY02, 08, 09, and 11, for subcellular localization?
5. In line 377, What's AMY's pfam number? Why isn't it introduced?
6. In line 394, The author introduced the phytozome v13 database for gene search, but the gene names in Table S2 were obviously not on this website, so the author needed to check them carefully.
7. In line 334, Previous literature indicated that there were 10 alpha-amylases in rice, but the author identified only 4 of them in Table S2. Based on these results, I doubt the accuracy of the author's gene family identification results, and the author needs to accurately identify the family genes of all the species in the article and redraw the corresponding pictures.
8. “Molecular characteristics and functional identification of the key alpha-amylase-encoding gene AMY11 in banana fruit” suggests replacing it with “Molecular characteristics and functional identification of a key alpha-amylase-encoding gene AMY11 in Musa acuminata”.
Author Response
Reviewer 1 Comments
Comments and Suggestions for Authors
The present study titled “Molecular characteristics and functional identification of the key alpha-amylase-encoding gene AMY11 in banana fruit” is presented in detailed well-elaborated research. In this paper, the alpha-amylase-encoding gene was systematically studied, and AMY11 was deeply analyzed and functionally identified. There are many mistakes in the article that need to be corrected and solved by the author, my comments related to improving the article are as follows:
- In figure 1c, The genetic difference between MaAMY11 and other families is very large, so I think it is necessary for the author to correct this gene and redraw it. And the green squares in Figure 1c should refer to the UTR region rather than upstream and downstream.
Answer: Thank you for your careful review and constructive comments on our manuscript. We carefully checked the MaAMY11 sequence and redraw Figure 1c in the revised manuscript, but the MaAMY11 is indeed large than the other members in banana and other plants. Thus, we added “…Interestingly, MaAMY11 has the longest amino acid sequence in the gene families compared with banana and other plants (Figure 1D and Figure S1)…” in the “Results” section of revised manuscript (Page 3 line 124-125). We added “…Notably, we found that MaAMY11 has the longest amino acid sequence compared with other plants (Figure S1). However, further research is necessary to determine whether MaAMY11 has a special function in multiple biological processes…” in the “Discussion” section of revised manuscript (Page 5 lines 233-236). We changed “…upstream and downstream…” into “…UTR…” in the “Figure 1c” of revised manuscript.
- In line 136-137 and 395-397, species names should be italicized.
Answer: Thank you. Species names have been changed to italics in the revised manuscript (Page 3 lines 128-130; Page 7 Lines 331-333).
- In figure 5, The clustering sequence of genes in the heat map of family gene expression is suggested to follow the sequence of evolutionary tree.
Answer: Thank you. We have redraw Figure 5 in the revised manuscript. But the clustering sequence of genes in the heat map of family gene expression is difficult to follow the sequence of evolutionary tree, so we removed the clustering analysis in the heat map.
- Why did the authors select only four genes, MaAMY02, 08, 09, and 11, for subcellular localization?
Answer: Thank you. In this study, we focused on MaAMY genes that are highly expressed at the transcriptional level in fruit organs or during fruit development and ripening. Among 12 MaAMYs, only four genes, including MaAMY02, 08, 09, and 11, were actively expressed (RPKM > 3.57) in fruit organs or during fruit development and ripening (Figure 5, Table S5 and S6), other eight members were not expressed (RPKM = 0) (Figure 5, Table S5 and S6). Therefore, we selected these four expressed genes for further subcellular localization.
- In line 377, What's AMY's pfam number? Why isn't it introduced?
Answer: Thank you. We changed “…ACDs were confirmed…” into “…ACDs (PF00128) were confirmed…” in the revised manuscript (Page 7 line 319).
- In line 394, The author introduced the phytozome v13 database for gene search, but the gene names in Table S2 were obviously not on this website, so the author needed to check them carefully.
Answer: Thank you. We have carefully re-checked the gene names in Table S3 of revised manuscript with the banana A genome database, phytozome v13 database, and ensembl plants database. We changed “…using the phytozome v13 database…” into “…using the banana A genome, phytozome v13 database and ensembl plants database…” in the revised manuscript (Page 7 lines 334-335).
- In line 334, Previous literature indicated that there were 10 alpha-amylases in rice, but the author identified only 4 of them in Table S2. Based on these results, I doubt the accuracy of the author's gene family identification results, and the author needs to accurately identify the family genes of all the species in the article and redraw the corresponding pictures.
Answer: Thank you. We accurately identify the family genes of all the species in the revised manuscript (Table S3) and redraw the corresponding Figure 2 and Figure 4. We changed “…a total of 156 AMY protein sequences from…” into “…a total of 108 AMY protein sequences from…” in the revised manuscript (Page 3 lines 127-128). A total of 10 alpha-amylases in rice were identified in the revised manuscript (Figure 2, 4 and Table S3).
- “Molecular characteristics and functional identification of the key alpha-amylase-encoding gene AMY11 in banana fruit” suggests replacing it with “Molecular characteristics and functional identification of a key alpha-amylase-encoding gene AMY11 in Musa acuminata”.
Answer: Thank you. We changed “…Molecular characteristics and functional identification of the key alpha-amylase-encoding gene AMY11 in banana fruit…” into “…Molecular characteristics and functional identification of a key alpha-amylase-encoding gene AMY11 in Musa acuminata…” in the “Title” of revised manuscript (Page 1 lines 2-3).
Reviewer 2 Report
Comments and Suggestions for Authors
The authors reported the genome-wide expression analysis of the alpha-amylase gene family in banana and the transient transformation of AMY11 in banana fruit. The manuscript contains interesting information. However, I have several comments and suggestions as below
General comments :
Please provide the raw data produced by the authors and stored in a public repository including the accession number. This information is essential.
The manuscript compares the promotors of all alpha-amylase gene families in this species and identifies the putative cis-element. However, the authors did not discuss is there any conserved cis-acting of this gene family promoter that is not found in other plant amylase promoters? Please discuss this result.
In addition, there is no information regarding promoter structure comparisons. Is there any 5 ’UTR’ introns of the promoters, TSS (transcription start site) location, etc ? This is essential to elaborate on the difference between gene expression level and specificity of gene expression, for instance, tissue-specific expression.
What is the particular reason that the authors does not take samples from stem organs? The authors should mention this also.
The manuscript data showed that MaAMY3 was not expressed in the fruit (Line 211) but in the introduction, it is shown that MaAMY3 is involved in starch degradation in banana fruit (Line 86) and from reference Xu et al. 2024. Please elaborate on this contradiction.
Interestingly, MaAMY11 have the longest amino acid in the gene families. Please elaborate this and compare with other plants.
Specific comments :
Figure 1A. What is the meaning of group I and II ?. What is the difference between group I, II and III from Figure 2A.
Figure 1C. Please explain what is Upstream/downstream in the legends.
I would like to recommend the authors draw and compare the amylase gene family based on known and predicted amino acid domains. For instance, transit peptide, carbohydrate bing module, and amylase catalytic, etc.
Figure 2B. Please revise Y-axis from “Ka/Ka” to “Ka/Ks” ratio.
Figure 3A. Please elaborate in the legend what is the color means.
Figure 3B. Please think again whether is it common or not to have 64, 104 or 144 TATA-box in the promoter region ?. I think this is a repeat region. Please elaborate again on this result.
Figure 4B and 4C. Please consider changing the picture, and comparing based on the monocots and dicots groups separated. Solanum lycopersicum and Arabidopsis thaliana (Dicots), in addition Oryza sativa and Zea mays (Monocots).
Figure 5A. Are you sure that the other MaAMYs were not expressed? Please include the housekeeping gene expression similar as qRT-PCR data (UBI and ACT) in this heatmap.
Figure 5C-J. Please put the gene name on top of the figure to avoid misunderstanding and it will be easier for the reader.
Please include the construct of the plasmid in the supplementary material.
Supplementary material Table S4. Has wrong column title should be DPF.
Supplementary material Table S5 was not really useful. It is enough to state in the manuscript.
Figure 7B. Please revise the figure. Figure 7B-E should be grouped as suppression. Figure 7-G-H should be grouped as overexpression.
Line 374. Please explain the abbreviation DAF and DPH. In this section. Also in the figure legend use this abbreviation.
Line 396. The species name is in italics. Please revise this.
Line 402. Explain that phylogenetics uses amino acid sequences or DNA sequences.
Line 436. Please provide a primer of the housekeeping gene used in this study.
Please include a reference for the software used in this study.
Comments on the Quality of English LanguageMinor editing of English language required
Author Response
Reviewer 2 Comments
Comments and Suggestions for Authors
The authors reported the genome-wide expression analysis of the alpha-amylase gene family in banana and the transient transformation of AMY11 in banana fruit. The manuscript contains interesting information. However, I have several comments and suggestions as below.
General comments:
- Please provide the raw data produced by the authors and stored in a public repository including the accession number. This information is essential.
Answer: Thank you. We provided raw data of Musa acuminata transcriptomic databases (SRX3938715, SRX3938722, SRX3938708, SRX3938709, SRX3938706, SRX3938707, SRX3938704). We changed “…Transcriptome data from various tissues and fruits at different developmental stages…” into “…Transcriptome data (SRX3938715, SRX3938722, SRX3938708, SRX3938709, SRX3938706, SRX3938707, and SRX3938704) from various tissues and fruits at different developmental stages…” in the revised manuscript (Page 4 lines 173-174).
- The manuscript compares the promotors of all alpha-amylase gene families in this species and identifies the putative cis-element. However, the authors did not discuss is there any conserved cis-acting of this gene family promoter that is not found in other plant amylase promoters? Please discuss this result.
Answer: Thank you. We added “…In addition, a large number of conserved cis-acting elements, including TATA-box and CAAT-box, were found in MaAMY gene promoter regions, which were consistent with the characteristics of the AMY gene promoters in rice [45] and cassava [10]…” in the “Discussion” section of revised manuscript (Page 6 lines 271-273).
- In addition, there is no information regarding promoter structure comparisons. Is there any 5 ’UTR introns of the promoters, TSS (transcription start site) location, etc? This is essential to elaborate on the difference between gene expression level and specificity of gene expression, for instance, tissue-specific expression.
Answer: Thank you. We added start site, end site, promoter regions, transcription start site (TSS) location, 5’untranslated region (UTR) introns, and score information regarding promoter structure comparisons in Table S4 of revised manuscript. We added “…promoter structure comparisons … database (http://www.fruitfly.org/seq_tools/promoter.html) and … software, respectively. These promoter regions didn’t contain 5’untranslated region (UTR) introns, all promoters contained transcription start sites (TSS). The location information of TSS was provided in Table S4…” in the revised manuscript (Page 4 lines 149-152).
- What is the particular reason that the authors does not take samples from stem organs? The authors should mention this also.
Answer: Thank you. The stem organs of banana were called as pseudostem, which was formed by overlapping leaf sheath. The leaves of banana include three parts: leaf sheaths, petioles, and leaves. The leaves and pseudostems of banana belong to homologous organs. Therefore, no stem organ was selected.
- The manuscript data showed that MaAMY3 was not expressed in the fruit (Line 211) but in the introduction, it is shown that MaAMY3 is involved in starch degradation in banana fruit (Line 86) and from reference Xu et al. 2024. Please elaborate on this contradiction.
Answer: Thank you. In the manuscript data, the MaAMY3 was named based on the order on the chromosomes, which was inconsistent with the naming convention used by Xu et al. (2024). We changed “…named MaAMY01–MaAMY12…” into “…named MaAMY01–MaAMY12 based on their order on the chromosomes…” in the revised manuscript (Page 3 line 99).
- Interestingly, MaAMY11 have the longest amino acid in the gene families. Please elaborate this and compare with other plants.
Answer: Thank you. We added “…Interestingly, MaAMY11 has the longest amino acid sequence in the gene families compared with banana and other plants (Figure 1D and Figure S1)…” in the “Results” section of revised manuscript (Page 3 line 124-125). We added “…Notably, we found that MaAMY11 has the longest amino acid sequence compared with other plants (Figure S1). However, further research is necessary to determine whether MaAMY11 has a special function in multiple biological processes…” in the “Discussion” section of revised manuscript (Page 5 lines 233-236).
Specific comments:
- Figure 1A. What is the meaning of group I and II ?. What is the difference between group I, II and III from Figure 2A.
Answer: Thank you. Group I and II in Figure 1A were the phylogenetic analysis of banana MaAMY protein family. Group I, II, and III from Figure 2A were the phylogenetic analysis of banana MaAMYs and other plants AMY amino acid sequences. We changed “…The AMYs were divided into three groups including group I, II, and III…” into “…The AMYs were divided into three groups including group I, II, and III based on amino acid sequences of AMYs in banana and other plants…” in the “Figure 2A” of revised manuscript (Page 13 lines 606-607).
- Figure 1C. Please explain what is Upstream/downstream in the legends.
Answer: Thank you. We changed “…Upstream/downstream…” into “…UTR…” in the “Figure 1C” of revised manuscript. We added “…UTR represents untranslated region. CDS represents coding sequence…” in the revised manuscript (Page 13 line 600).
- I would like to recommend the authors draw and compare the amylase gene family based on known and predicted amino acid domains. For instance, transit peptide, carbohydrate bind module, and amylase catalytic, etc.
Answer: Thank you. We added “…a alpha-amylase C-terminal domain or a signal peptide was found in MaAMY02, 04, 05, 06, 08, 09, 10, 11, and 12 (Table S2). Interestingly, MaAMY11 has the longest amino acid sequence in the gene families compared with banana and other plants (Figure 1D and Figure S1)…” in the “Results” section of revised manuscript (Page3 lines 122-125).
- Figure 2B. Please revise Y-axis from “Ka/Ka” to “Ka/Ks” ratio.
Answer: Thank you. We changed “Ka/Ka ratio” into “Ka/Ks ratio” in the “Figure 2B” of revised manuscript.
- Figure 3A. Please elaborate in the legend what is the color means.
Answer: Thank you. We added “…Different colors represent different gene density on the chromosome. Green, blue, and orange represent low, medium, and high gene density, respectively…” in the “Figure 3A” of revised manuscript (Page 13 lines 614-616).
- Figure 3B. Please think again whether is it common or not to have 64, 104 or 144 TATA-box in the promoter region ?. I think this is a repeat region. Please elaborate again on this result.
Answer: Thank you. We changed “…TATA-box and CAAT-box…” into “…TATA-box (9-144) and CAAT-box (8-20)…” in the “Results” section of revised manuscript (Page 4 line 153). We added “…In addition, a large number of conserved cis-acting elements, including TATA-box and CAAT-box, were found in MaAMY gene promoter regions, which were consistent with the characteristics of the AMY gene promoters in rice [45] and cassava [10]…” in the “Discussion” section of revised manuscript (Page 6 lines 271-273).
- Figure 4B and 4C. Please consider changing the picture, and comparing based on the monocots and dicots groups separated. Solanum lycopersicum and Arabidopsis thaliana (Dicots), in addition Oryza sativa and Zea mays (Monocots).
Answer: Thank you. We changed the Figure 4B and 4C in revised manuscript, and compared based on the monocots (Oryza sativa and Zea mays) and dicots (Solanum lycopersicum and Arabidopsis thaliana) groups separated.
- Figure 5A. Are you sure that the other MaAMYs were not expressed? Please include the housekeeping gene expression similar as qRT-PCR data (UBI and ACT) in this heatmap.
Answer: Thank you. We included the housekeeping gene (UBQ2 and Actin) expression in Figure 5A-B.
- Figure 5C-J. Please put the gene name on top of the figure to avoid misunderstanding and it will be easier for the reader.
Answer: Thank you. We put the gene name on top of the figure 5C-J.
- Please include the construct of the plasmid in the supplementary material.
Answer: Thank you. We included the construct of the plasmid in the Figure S2.
- Supplementary material Table S4. Has wrong column title should be DPF.
Answer: Thank you. We changed “DPH” into “DAF” in Table S6 of revised manuscript.
- Supplementary material Table S5 was not really useful. It is enough to state in the manuscript.
Answer: Thank you. We deleted Table S5 in the revised manuscript.
- Figure 7B. Please revise the figure. Figure 7B-E should be grouped as suppression. Figure 7-G-H should be grouped as overexpression.
Answer: Thank you. We revised the Figure 7. Figure 7B-E was grouped as suppression. Figure 7G-H was grouped as overexpression.
- Line 374. Please explain the abbreviation DAF and DPH. In this section. Also in the figure legend use this abbreviation.
Answer: Thank you. We changed “…DAF…” into “…days after the emergence of the inflorescence from the pseudostem (DAF)…” in the revised manuscript (Page 7 line 312). We changed “…DPH…” into “…days postharvest (DPH)…” in the revised manuscript (Page 7 line 314).
- Line 396. The species name is in italics. Please revise this.
Answer: Thank you. We have revised the species name to italics in the revised manuscript (Page 3 lines 128-130; Page 7 lines 332-335).
- Line 402. Explain that phylogenetics uses amino acid sequences or DNA sequences.
Answer: Thank you. We changed “…AMYs…” into “…AMY amino acid sequences…” in the revised manuscript (Page 8 line 339).
- Line 436. Please provide a primer of the housekeeping gene used in this study.
Answer: Thank you. We provide a primer of the housekeeping gene in Table S8 of revised manuscript. We added “…Table S8 provided the primer sequence information…” in the revised manuscript (Page 8 lines 371-372).
- Please include a reference for the software used in this study.
Answer: Thank you. All software and database used in this study were listed in the Table S7.
- Minor editing of English language required
Answer: Thank you. We revised the WHOLE manuscript carefully and improved English readability. In addition, we have carefully proofread the entire manuscript to ensure precision and accuracy.
Reviewer 3 Report
Comments and Suggestions for Authors
Some very minor edits suggested in the returned ms
comparing banana and 'other' cereals doesn't seem correct; both are monocots but banana isn't a cereal.

Author Response
Reviewer 3
- Some very minor edits suggested in the returned ms.
Answer: Thank you. We changed “…the transcription factor…” into “…it was shown that the transcription factor…” in the revised manuscript (Page 2 line 82).
- ie amino acids-why mentions.
Answer: Thank you. The purpose of mentioning amino acids in the revised manuscript (Page 3 lines 103-109) is to compare the similarities and differences in amino acid composition, structure, and physicochemical characteristics among 12 MaAMY members, laying the foundation for the functions of different MaAMY members.
- Is banana considered a cereal? comparing banana and 'other' cereals doesn't seem correct; both are monocots but banana isn't a cereal.
Answer: Thank you. We changed “…monocot cereals…” into “…monocot plants…” in the revised manuscript (Page 3 lines 135-137).
- “…which may be a response to high expression at the postharvest ripening stages, implying a potential regulatory function…”. Cause?
Answer: Thank you. We changed “…which may be a response to high expression at the postharvest ripening stages, implying a potential regulatory function…” into “…which may be play an important role in the phytohormone- and development-related biological processes…” in the revised manuscript (Page 6 lines 270-271).
Round 2
Reviewer 1 Report
Comments and Suggestions for Authors
The authors have addressed all my comments and accordingly revised the manuscript. I have no further comment on the current version of manuscript.
Reviewer 2 Report
Comments and Suggestions for Authors
The revised manuscript is improved. All of the comments have been addressed. No additional comments.